# Wind-Driven Coastal Upwelling near Large River Deltas in the Laptev and East-Siberian Seas

**Alexander Osadchiev [1,2,*]** , **Ksenia Silvestrova [1]** and **Stanislav Myslenkov [1,3,4]**

[1] Shirshov Institute of Oceanology, Russian Academy of Sciences, Nakhimovskiy prospect 36, 117997 Moscow, Russia; silvestrova.kp@ocean.ru (K.S.); myslenkov@nral.org (S.M.)

[2] Institute of Geology of Ore Deposits, Petrography, Mineralogy and Geochemistry, Russian Academy of Sciences, Staromonetny pereulok 35s2, 119017 Moscow, Russia

[3] Faculty of Geography, Moscow State University, Leninskie Gory 1, 119992 Moscow, Russia

[4] Marine Forecast Division, Hydrometeorological Research Centre of the Russian Federation, B. Predtechenskiy pereulok 11-13, 123242 Moscow, Russia

* Correspondence: osadchiev@ocean.ru

**Abstract:** The Lena, Kolyma, and Indigirka rivers are among the largest rivers that inflow to the Arctic Ocean. Their discharges form a freshened surface water mass over a wide area in the Laptev and East-Siberian seas and govern many local physical, geochemical, and biological processes. In this study we report coastal upwelling events that are regularly manifested on satellite imagery by increased sea surface turbidity and decreased sea surface temperature at certain areas adjacent to the Lena Delta in the Laptev Sea and the Kolyma and Indigirka deltas in the East-Siberian Sea. These events are formed under strong easterly and southeasterly wind forcing and are estimated to occur during up to 10%–30% of ice-free periods at the study region. Coastal upwelling events induce intense mixing of the Lena, Kolyma, and Indigirka plumes with subjacent saline sea. These plumes are significantly transformed and diluted while spreading over the upwelling areas; therefore, their salinity and depths abruptly increase, while stratification abruptly decreases in the vicinity of their sources. This feature strongly affects the structure of the freshened surface layer during ice-free periods and, therefore, influences circulation, ice formation, and many other processes at the Laptev and East-Siberian seas.

**Keywords:** coastal upwelling; wind forcing; river plume; MODIS; Arctic Ocean

## 1. Introduction

The Arctic Ocean covers an area of about 3% of the World Ocean area and holds only 1% of its volume, but receives approximately 11% of world continental discharge [1,2]. This enormously large freshwater runoff forms large freshened water masses at the Arctic shelf and induces strong vertical stratification that plays a crucial role in the variability of ice cover and regional albedo [3–5]. As a result, the spreading and mixing of freshwater runoff in the Arctic Ocean influences global climate processes. Freshened water masses also significantly affect many local processes in the Arctic Ocean, especially in coastal and shelf areas where the impact of freshwater discharge is the strongest [6–14].

The Lena, Kolyma, and Indigirka rivers are among the largest rivers that inflow to the Arctic Ocean. Annual discharges of the Lena, Kolyma, and Indigirka rivers are estimated as 530, 130, and 60 km$^3$ and they provide approximately 70% and 75% of the total freshwater discharge to the Laptev and East-Siberian seas, respectively [15,16]. The majority of this discharge inflows to the sea during the ice-free period in June–September and forms the Lena, Kolyma, and Indigirka river plumes [17]. These buoyant plumes occupy hundreds of thousands square kilometers in the Laptev and East-Siberian

seas and are among the largest freshwater reservoirs in the Arctic Ocean [17–20]. Spreading and transformation of these river plumes determine vertical stratification and, therefore, strongly affect circulation and ice formation in the Laptev and East-Siberian seas, as well as many other physical, geochemical, and biological processes [21–32].

In this study we focus on upwelling events which regularly occur at coastal areas adjacent to the deltas of the Lena, Kolyma, and Indigirka rivers. Surface manifestations of these upwelling events are visible on ocean color satellite imagery due to elevated turbidity and on sea surface temperature (SST) satellite imagery due to reduced temperature. However, correct identification of the origin of SST and turbidity features observed on satellite imagery is not a straightforward task. SST features in the study region are formed as a result of interaction between water masses with different temperature, namely, warm river plumes and cold saline sea water, and are associated with spreading of river plumes, mixing of surface layer with subjacent sea, and ice melting. Areas of elevated sea surface turbidity in coastal and shelf regions are commonly associated with four different processes: spreading of turbid river plumes, coastal erosion, resuspension of bottom sediments penetrated to sea surface, and algal blooms [33]. The first three processes are common features of the Laptev and East-Siberian seas [25,27,34,35], while algal blooms do not occur in these seas [32,36–38]. Turbid regions associated with river plumes are adjacent to river estuaries and deltas. Spatial and temporal variability of these regions is defined mainly by river discharge rate, turbidity of river water, and local wind forcing [39–47]. Coastal erosion in the Laptev and East-Siberian seas is extremely intense due to active thermal abrasion. It provides large land-ocean fluxes of terrigenous sediments whereby total eroded sediment volume exceeds river sediment discharge [31,33,48]. Turbid regions associated with coastal erosion are adjacent to long segments of sea coast, but it does not cause elevated turbidity in offshore areas. Finally, resuspension of bottom sediments occurs in shallow areas and can be caused by upwelling events, tides, and wind waves [24,49–53]. In the latter case, turbulence induced by breaking surface waves penetrates from the surface layer to the sea bottom, causing resuspension of bottom sediments and their subsequent upward convection to surface layer. Tidal circulation and coastal upwelling, conversely, initially induce turbulence in the bottom layer, which penetrates upward and can reach the surface layer carrying resuspended sediments.

Interaction between river plumes and coastal upwelling were addressed in many previous works. Stratification in the coastal area affects the depth of the mixed layer and alters wind-driven cross-shore circulation [54–57]. Upwelling-favorable winds induce offshore transport of river plumes and their detachment from the sea shore [58–62]. As a result, a sharp salinity gradient is formed between the saline and low-stratified near-shore area and offshore located river plume [36]. Upwelling winds also cause intense mixing of a river plume with subjacent saline sea due to increased velocity shear and an Ekman straining mechanism [58,63]. Therefore, upwelling events along coastal areas influenced by freshwater discharge significantly affect spreading and mixing of river plumes, as well as the local nutrient cycle, biological consumption, food webs, and biological productivity [56,64–68].

Many previous works addressed wind-driven coastal upwelling events in the Arctic Ocean [19,69–74]. However, interaction between river plumes and coastal upwelling in the Arctic Ocean remain mainly unstudied. We are aware of only a few studies focused on coastal upwelling influenced by large freshwater discharge, namely, the Mackenzie River [75–78]. In this study we report coastal upwelling events that occur over wide areas adjacent to deltas of the Lena, Indigirka, and Kolyma rivers. Using satellite imagery and atmospheric reanalysis fields, we reveal that these upwelling events are regularly induced by wind forcing. We show that they strongly affect the thermohaline and turbidity properties of the sea surface layer and influence spreading and mixing of the large Lena, Kolyma, and Indigirka plumes.

The paper is organized as follows. Section 2 provides detailed information about the study area, the satellite and wind reanalysis data, and the methods of detection of upwelling events used in this study. Section 3 describes spatial and temporal characteristics of wind-driven coastal upwelling events that occur near the Lena, Kolyma, and Indigirka deltas. Frequency and duration of these coastal

upwelling events are assessed and their influence on the spreading and mixing of the Lena, Kolyma, and Indigirka plumes is analyzed in Section 4, followed by the conclusions in Section 5.

## 2. Study Area, Data, and Methods

### 2.1. Study Area

The Laptev and East-Siberian seas are located at the east of the Eurasian part of the Arctic Ocean. These seas are semi-enclosed by the Siberian coast and large archipelagos and islands (Severnaya Zemlya, New Siberian Islands, and Wrangel Island) in the south, east, and west, and only in the north they are open to the central part of the Arctic Ocean (Figure 1). Half of the Laptev Sea and almost the whole area of the East-Siberian Sea rest on the continental shelf. The distance between the sea shore and the continental slope increases from 100 to 200 km at the western part of the Laptev Sea and to 1000 km at the eastern part of the East-Siberian Sea. Average sea depths of the Laptev and East-Siberian seas are 580 and 45 m, respectively.

General circulation in the Laptev and East-Siberian seas is governed by river runoff and zonal water exchange with the Kara Sea [79], the Chukchi Sea [80], and the deep basin of the Arctic Ocean [12]. The Laptev and East-Siberian seas receive a large volume of continental discharge, approximately 800 km$^3$ to the Laptev Sea and 250 km$^3$ to the East-Siberian Sea annually, which accounts for approximately a quarter of the total freshwater runoff to the Arctic Ocean [1,15,81]. Spatial and temporal variability of river plumes formed in these seas are mainly governed by river discharge rates and wind forcing conditions [21,23,25,27,80–83]. Tidal circulation in the Laptev and East-Siberian seas is dominated by a lunar semidiurnal tidal wave that propagates from the North Atlantic to the Arctic Ocean. Tidal amplitudes in these seas are generally low, as compared to the World Ocean, and do not exceed 0.5 m in the majority of their area [84–86].

The Laptev and East-Siberian seas are frozen during the majority of a year. The southern parts of the seas adjacent to the Lena, Indigirka, and Kolyma deltas are covered by landfast ice (1.5–2 m thick) from the end of October to June–July. The ice regime in the study areas is significantly influenced by continental runoff [17,25,27] and the Great Siberian Polynya [87]. Summer and autumn ice coverage of these seas shows large inter-annual variability. The northernmost position of the edge of the sea ice was located at a distance of 200–300 km from the Siberian shore during certain years (e.g., 2013, 2014, 2018), so the central and northern parts of these seas were covered by ice during the whole year. On the other hand, these seas can be totally free of ice at the end of August–beginning of October during the years of reduced ice coverage (e.g., 2012, 2017, 2019).

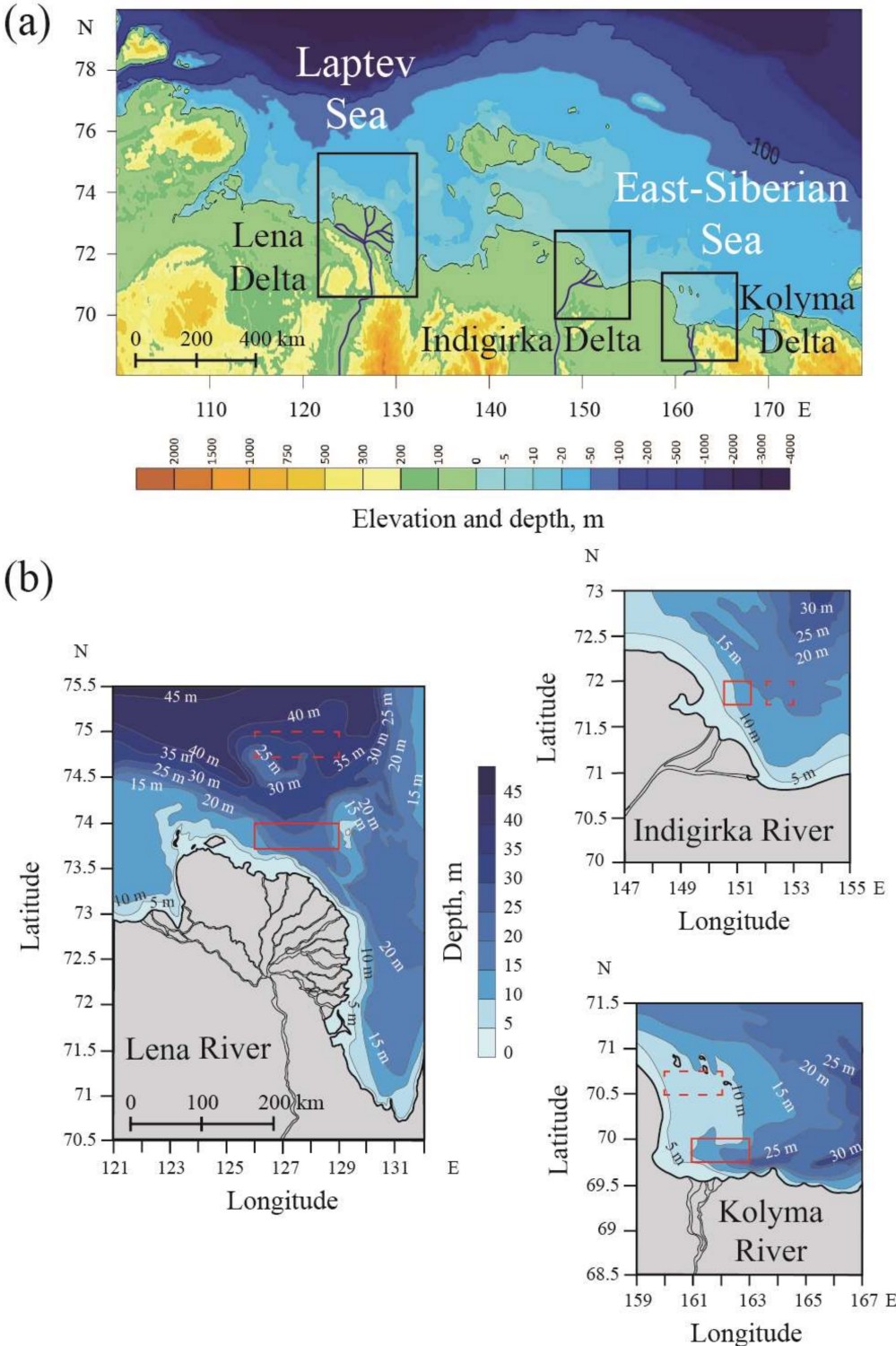

**Figure 1.** (**a**) Bathymetry and topography of the study region locations of the Lena, Indigirka, and Kolyma deltas in the Laptev and East-Siberian seas; (**b**) bathymetry of the areas adjacent to the Lena, Indigirka, and Kolyma deltas. The graphic scales correspond to the latitude of 72°. Red boxes indicated in panel (**b**) show locations of reference areas in the upwelling regions (dashed contours) and the ambient sea (solid contours) used to identify upwelling events.

*2.2. Data and Methods*

Satellite data used in this study include Terra/Aqua Moderate Resolution Imaging Spectroradiometer (MODIS) satellite imagery for the period 2000–2019 provided by the National Aeronautics and Space Administration (NASA). MODIS L1b calibrated radiances including MODIS bands 1 (red), 3 (blue), 4 (green), and daytime 31 (thermal) were downloaded from the NASA web repository (https://ladsweb.modaps.eosdis.nasa.gov/). We used ESA BEAM software for retrieving maps of sea surface distributions of corrected reflectance and brightness temperature at the study areas with spatial resolutions of 100 m and 1 km, respectively. Wind forcing conditions were examined using NCEP/CFSR/CFsv2 atmospheric reanalysis with a ~0.3° (1979–2010) and ~0.2° (2011–2019) spatial and hourly temporal resolution [88,89]. The reanalysis data were downloaded from the National Climatic Data Center of the National Oceanic and Atmospheric Administration (NCDC NOAA) web repository (https://www.ncdc.noaa.gov/). Visual inspection of all satellite images of three study areas (Figure 1b) acquired during ice-free seasons (July–October) of 2000–2019 was performed to detect cloud-free and ice-free satellite images. The resulting 252 images were used to detect upwelling events near the Lena, Indigirka, and Kolyma deltas in the following way. For every considered region we identified a pair of reference areas, namely, the upwelling area adjacent to the delta and the ambient sea area not affected by upwelling events. The pairs of these reference areas are shown in Figure 1b by red boxes, while their coordinates are given in Table 1. Then for every cloud-free and ice-free satellite image we calculated differences in average brightness temperature within the pairs of reference areas. If the temperature of an upwelling area was smaller than the temperature of an ambient sea area by >2 °C, we regarded this case as a "cold event" bounded by a "distinct" frontal zone which is a surface manifestation of upwelling.

**Table 1.** Coordinates of reference areas used to identify upwelling events near the Lena, Indigirka, and Kolyma deltas.

|  | Lena Delta | | Indigirka Delta | | Kolyma Delta | |
|---|---|---|---|---|---|---|
|  | Upwelling Area | Ambient Sea | Upwelling Area | Ambient Sea | Upwelling Area | Ambient Sea |
| Longitude, °E | 126–129 | 126–129 | 150.5–151.5 | 152–153 | 161–163 | 160–162 |
| Latitude, °N | 73.75–74 | 74.75–75 | 71.75–72 | 71.75–72 | 69.75–70 | 70.5–70.75 |

Due to the complexity of coastal processes that govern the temperature of the sea surface and the absence of specific regional algorithms for retrieving SST in the study areas with very limited in situ measurements, we did not used the standard SST product of MODIS. Instead, we used a brightness temperature product that does not provide an accurate temperature of the sea surface, but shows relative temperature differences, which can be used to detect upwelling events. The qualitative routine for detection of upwelling events was based on the brightness temperature values, which was followed by assessment of surface turbidity during upwelling and non-upwelling events. Due to the absence of specific regional algorithms for retrieving total suspended matter in the study area influenced by multiple processes (resuspension of bottom sediments, river discharge, coastal erosion), we did not apply quantitative assessment of surface turbidity, but performed qualitative visual inspection that identified elevated turbidity at the upwelling area in all cases during and shortly after upwelling events and mainly normal turbidity during non-upwelling periods. As a result, hereafter in the text, we regard "cold events" as "cold and turbid events".

## 3. Results

*3.1. Coastal Upwelling near the Lena Delta in the Laptev Sea*

Optical satellite imagery regularly reveals events of increased sea surface turbidity and reduced sea surface temperature at the area located to the north from the Lena Delta (Figure 2). To study this

feature, we analyzed all MODIS Terra and MODIS Aqua satellite images of the study region taken in 2000–2019 during July–October when the southern part of the Laptev Sea was free of ice. Due to common cloudy weather conditions, we detected only 25 periods (1–6 days long) when the area adjacent to the Lena Delta was clearly seen in optical satellite images and the structure of surface turbidity and temperature could be identified. Cold and turbid sea to the north from the Lena Delta was observed in 12 cases of the 25 considered periods. The other 13 cases were characterized by relatively homogenous turbidity and temperature at the study area, without any distinct frontal zones.

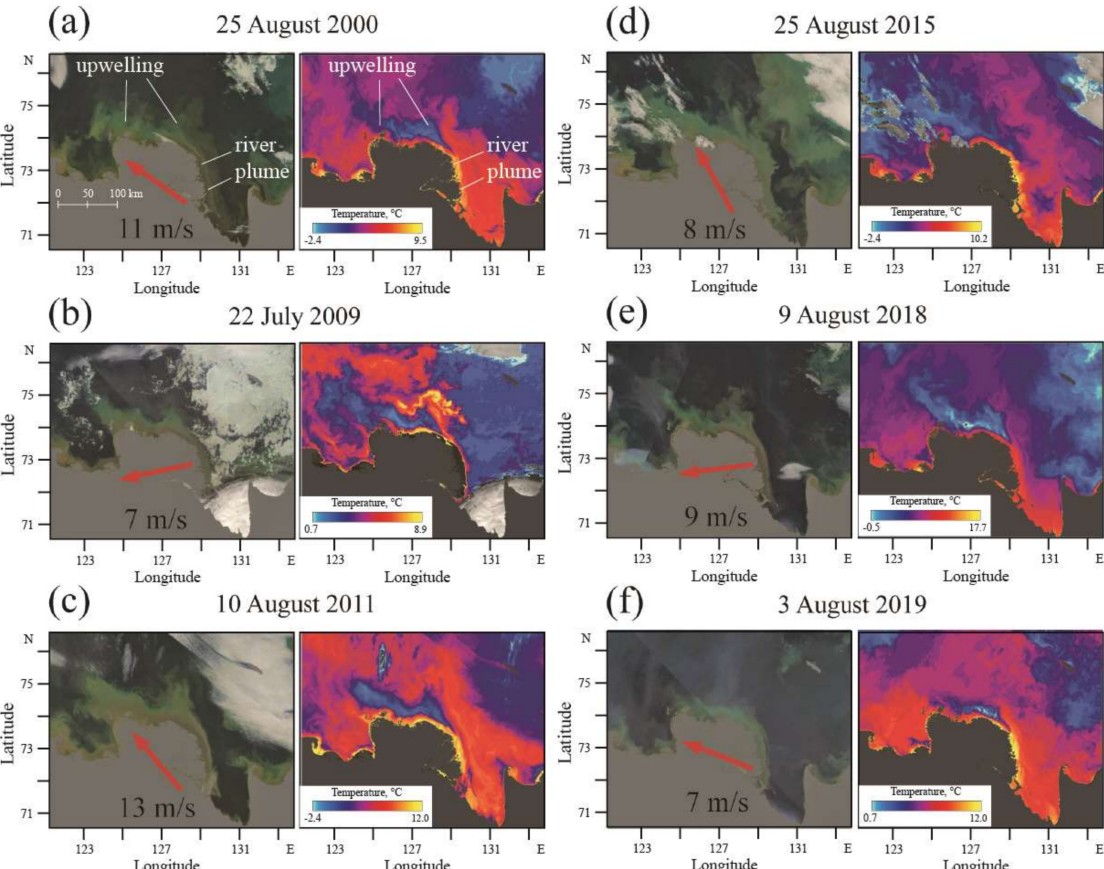

**Figure 2.** Corrected reflectance (left) and brightness temperature (right) from MODIS (Moderate Resolution Imaging Spectroradiometer) Terra and MODIS Aqua satellite images of the Laptev Sea acquired on (**a**) 25 August 2000, (**b**) 22 July 2009, (**c**) 10 August 2011, (**d**) 25 August 2015, (**e**) 9 August 2018, and (**f**) 3 August 2019 indicating the location of upwelling events to the north of the Lena Delta induced by wind forcing (arrows) and manifested by elevated sea surface turbidity and reduced sea surface temperature. Surface manifestations of upwelling events and river plumes are indicated in panel (**a**).

Typical examples of the cold and turbid events observed during six different days in 2000–2019 are shown in Figure 2. Sharp sea surface temperature gradients are formed between the cold area located to the north from the Lena Delta and the surrounding warm sea. This cold area is bounded by the distinct frontal zone whose location and shape is stable on satellite images taken on different days. The location and shape of this thermal frontal zone show good agreement with local bathymetry (Figure 3, right panels). The southern and eastern parts of the thermal frontal zone are located over the isobaths of 5–10 m stretched along the northern coast of the Lena Delta and the large shoal located to the northeast from the Lena Delta. The northern part of the thermal frontal zone is generally located over the isobath of 30 m, but its position was less stable. The cold area typically occupies a large part of the coastal sea (15,000–17,000 km$^2$), apart from two days, 11 September 2005 and 2–4 August 2019,

when this area was relatively small and the northern part of the thermal frontal zone had shifted in a southeasterly direction (Figure 3a,b, right panels).

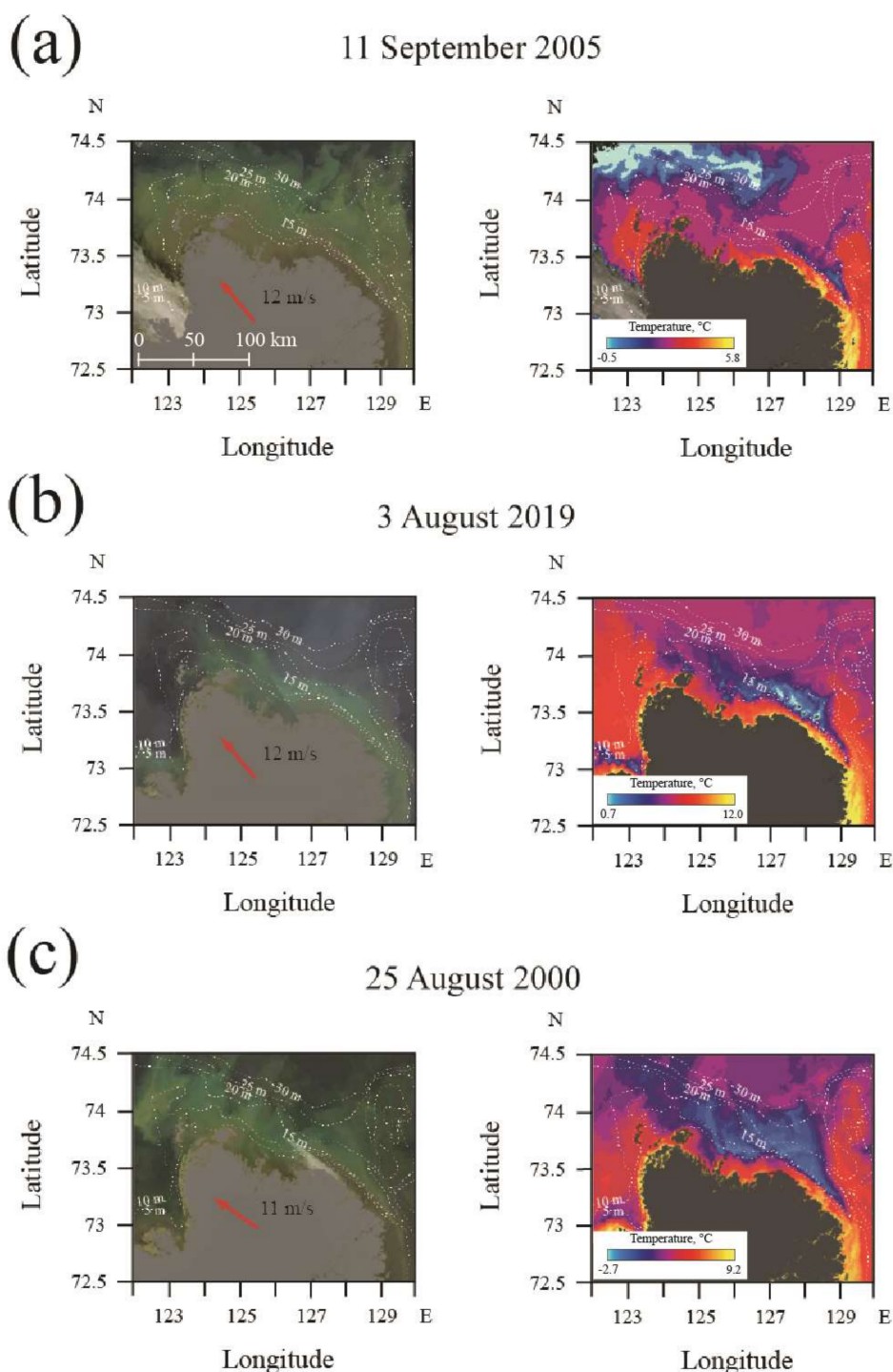

**Figure 3.** Corrected reflectance (left) and brightness temperature (right) from MODIS Terra and MODIS Aqua satellite images of the area adjacent to the Lena Delta acquired on (**a**) 11 September 2005, (**b**) 3 August 2019, and (**c**) 25 August 2000 illustrating initial (**a**), middle (**b**), and well-developed (**c**) stages of formation of upwelling events in response to wind forcing (arrows).

The surface turbidity structure of the study region during the cold and turbid events was more complex than the surface temperature structure. Surface turbidity was elevated to the north from the Lena Delta at the area occupied by cold surface water (Figure 2). Elevated turbidity was also registered along the eastern part of the Lena Delta. We associate elevated turbidity to the north of the delta with upwelling events and elevated turbidity along the eastern part of the delta with the Lena plume, due to the following reasons. Around 80–90% of freshwater and sediment discharge of the Lena River inflows to the Laptev Sea from the eastern part of the Lena Delta, while its northern part accounts only for 5–8% [90]. As a result, a large turbid and warm river plume is formed only along the eastern part of the Lena Delta. Therefore, areas of elevated turbidity located to the north from the Lena Delta are not likely to be formed by turbid river discharge.

As was described in Section 1, discharge-induced, erosion-induced, and resuspension-induced turbidity events can have similar sea surface manifestations on optical satellite imagery that hinders detection of their origin. However, these processes can be distinguished using other characteristics of sea water. River plumes generally have different salinity, temperature, concentrations of chlorophyll a and dissolved organic matter, as compared to adjacent sea water [91–94]. In particular, river plumes formed in the Laptev and East-Siberian seas during summer and early autumn are significantly warmer than surrounding sea due to the large temperature difference between river and sea water [17,25,27]. The surface temperature in sea areas influenced by coastal erosion is also greater than in surrounding sea due to the absorption of heat from sunlight by suspended particles in the absence of vertical convection. Sea areas influenced by bottom resuspension, on the other hand, are colder than the surrounding sea during the warm season due to mixing of the warmer surface layer with colder bottom water. As a result, the cold and turbid zones observed to the north of the Lena Delta are caused by bottom resuspension, while warm and turbid zones along the coast of Lena Delta are caused by spreading of turbid river plumes.

As was discussed in Section 1, upwelling events, tides, and wind waves are the three possible processes that induce bottom resuspension and form the considered cold and turbid zone. Tidal circulation is very low in the central part of the Laptev Sea and limitedly affects mixing at the study area [62,65]. Coastal upwelling events are commonly manifested by cold and turbid zones in satellite imagery in many world regions [95–98]. Distributions of sea surface temperature observed during coastal upwelling events show significant dependence on local bathymetry. Shapes of cold surface areas formed by upwelling are consistent with isobaths and cores of cold areas are commonly detached from the sea shore [99]. This is the case of the cold and turbid area observed to the north of the Lena Delta which is stably located between isobaths of 5–10 and 30 m. Therefore, this area is not formed as a result of mixing by wind waves, because this process does not depend on bathymetry and can cause surface mixing over both shallow and deep sea areas.

Wind forcing in the study area obtained from the NCEP/CFSR/CFSv2 wind reanalysis confirms that the cold and turbid zone to the north of the Lena Delta is formed by wind-driven upwelling events. Figure 4 shows wind direction (measured in a clockwise direction from north) and wind speed at the area of formation of cold and turbid events. Daily averaged wind forcing conditions are shown in Figure 4 for the days of satellite observations (filled symbols) and for the preceding days (empty symbols). All cold and turbid events detected on satellite imagery (red squares) occurred either during strong southeast winds or shortly after their secession, i.e., average wind speed exceeded 8 m/s and average wind direction was between 120° and 170° on the day of satellite observation or on the preceding day. In particular, if a filled red square is located outside the black dashed rectangle (indicating the upwelling-favorable conditions) in Figure 4, its corresponding empty red square is located inside the dashed rectangle. Therefore, we consider these events as residual upwelling events, i.e., upward penetration of cold and turbid water that did not dissipate shortly after secession of an upwelling wind. On the other hand, there were no cases when an absence of a cold and turbid zone occurred during (filled blue triangles) or shortly after (empty blue triangles) strong upwelling winds were detected, i.e., all triangles in Figure 4 are located outside the black dashed rectangle. Figure 4

shows an asymmetry in wind conditions with almost absent wind forcing between 135° and 225°. This feature is presumably caused by the dependence of cloud coverage of the considered coastal areas on wind direction. Offshore areas of the Laptev Sea are mostly constantly covered by clouds due to intense evaporation, while the land is often cloud-free. As a result, wind that blows from sea to land induces the transport of clouds from the open sea to coastal areas, which hinders optical satellite observations of the sea surface. As a result, there are almost no wind forcing conditions between 135° and 225° among the relatively small sets of cloud-free satellite images of the considered deltaic area.

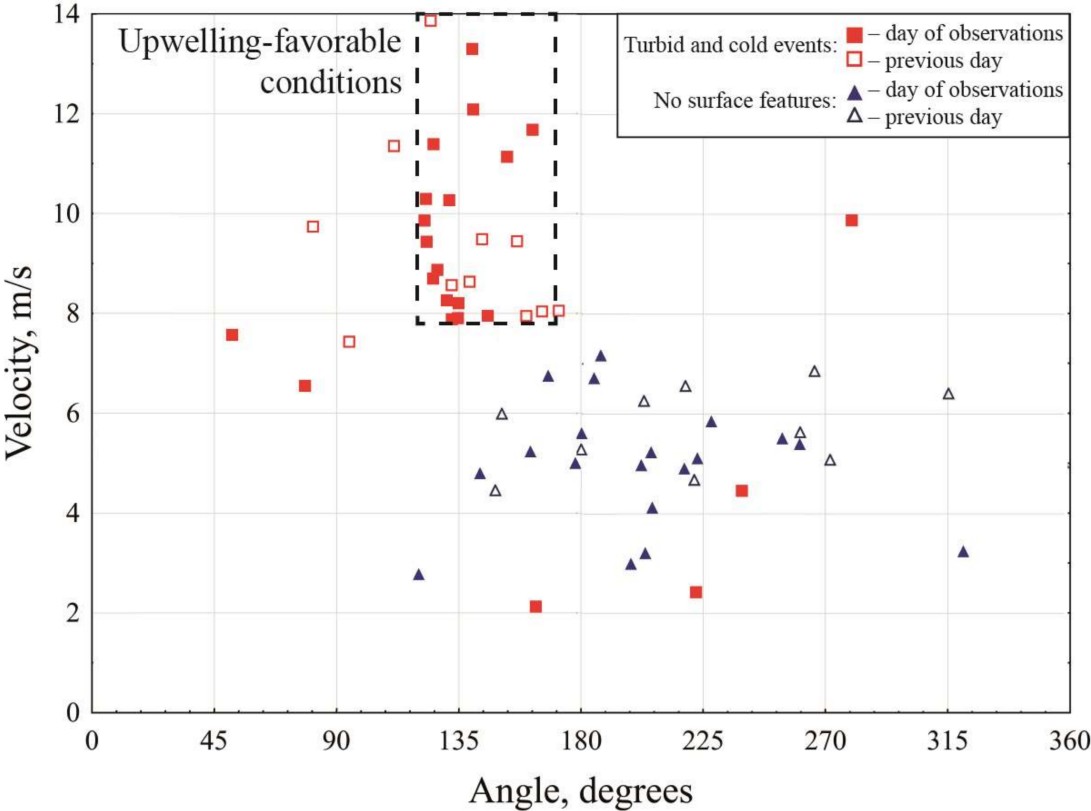

**Figure 4.** Wind forcing conditions at the Lena Delta region during periods of presence (red squares) and absence (blue triangles) of cold and turbid events detected on satellite imagery. For each satellite image, averaged wind forcing conditions are shown during the day of satellite observation (filled symbols) and during the preceding day (empty symbols). The black dashed rectangle indicates upwelling-favorable wind forcing conditions.

Joint analysis of satellite imagery and wind forcing conditions revealed different stages of formation and dissipation of upwelling events in response to changes of wind forcing regimes (Figures 3 and 5). Early stage of formation of upwelling events characterized by a small area of the cold and turbid zone was detected twice, namely, on 9–11 September 2005 and 2–4 August 2019. South wind forcing was prevailing in the study region on 8–10 September 2005 and changed its direction to southeast (12 m/s) on 11 September 2005. No upwelling manifestations were observed on satellite images acquired on 9 and 10 September 2005. On the next day, 11 September 2011, a relatively small cold area was detected at the isobaths of 5–10 m between the northeastern coast of the Lena Delta and the large shoal, indicating the beginning of formation of the upwelling event (Figure 3a, right panel). Large cold area located northwestward from the Lena Delta on 11 September 2005 was formed by ice melting and does not relate to vertical mixing processes. Satellite imagery acquired on 2–4 August 2019 shows the development of the upwelling event in response to strong southeast wind (7–8 m/s), which started dominating in the study region on 1 August 2019. As on 11 September 2011, the cold and



turbid upwelling zone initially was formed at the northwestward coast of the Lena Delta and steadily propagated westward along the isobaths of 5–10 m and then northward towards the isobath of 30 m. The upwelling area steadily increased from 2000 km$^2$ on 2 August to 4500 km$^2$ on 4 August (Figure 3b).

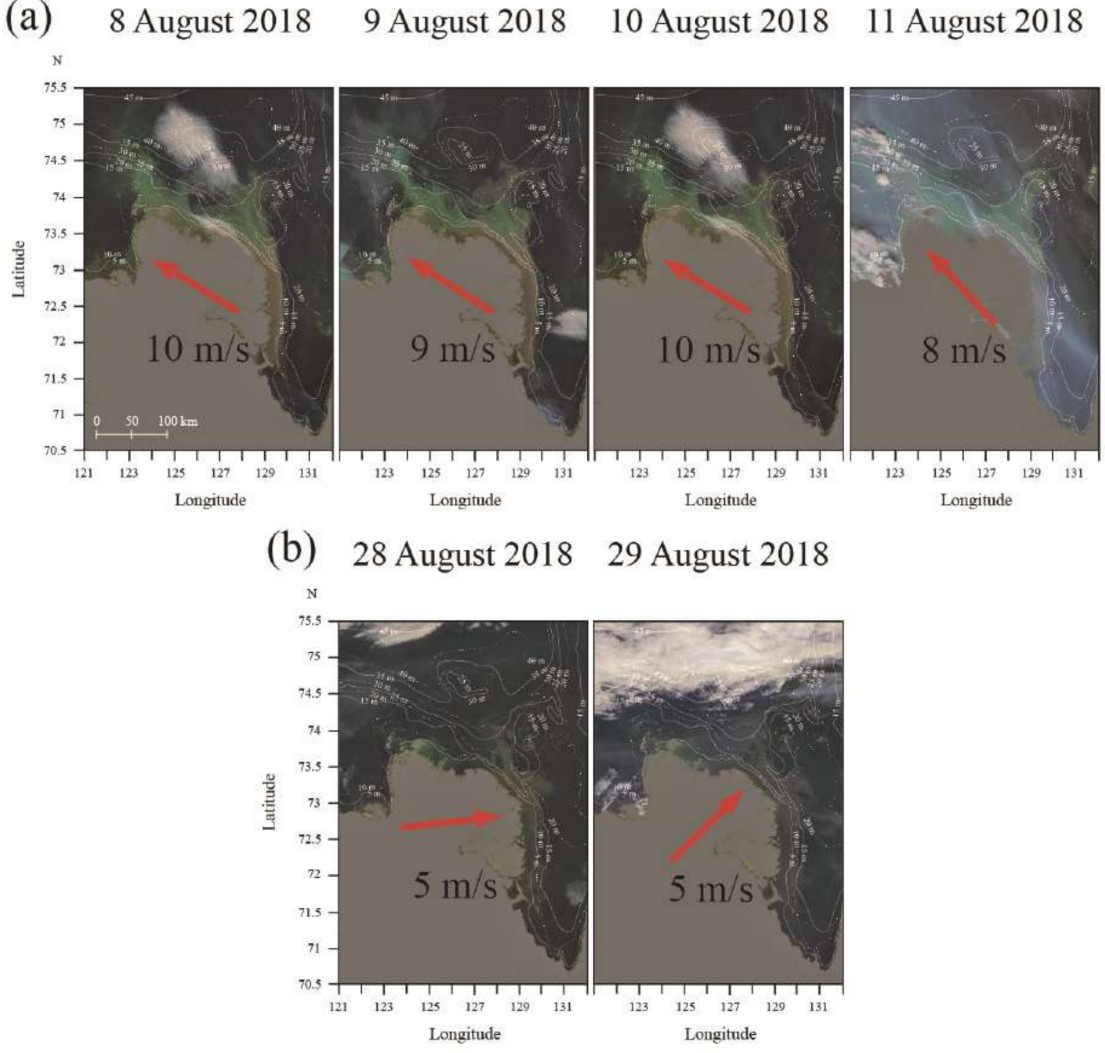

**Figure 5.** Corrected reflectance from MODIS Terra and MODIS Aqua satellite images of the area adjacent to the Lena Delta acquired on (**a**) 8–11 August 2018, and (**b**) 28–29 August 2018 and wind forcing (arrows) during upwelling (**a**) and non-upwelling (**b**) events.

Well-developed upwelling events that resulted in formation of a cold and turbid zone up to the isobath of 30 m were registered after 4–5 days of upwelling winds. In particular, this case was observed on 25 August 2000 after 4 days of strong southeasterly wind (7–11 m/s) (Figure 3c). After the development of an upwelling event, the cold and turbid zone remained stable and did not spread offshore. Satellite images acquired on 5, 6, 8, 9, 10, and 11 August 2018 during upwelling wind forcing showed that the area of the fully developed upwelling zone did not change (Figure 5a). However, after secession of upwelling wind, this cold and turbid area dissipated and was not observed on satellite imagery acquired on 28–29 August 2018 (Figure 5b). Steady dissipation of the cold and turbid area was also registered at the end of August 2015. A week of strong easterly winds on 15–23 August 2015 caused formation of an upwelling event, whereby surface manifestation bounded by a distinct frontal zone was observed on satellite imagery acquired on 24 August 2015. After secession of upwelling winds on 24 August 2015, sharp temperature and turbidity gradients between the upwelling zone and the adjacent sea steadily dissipated. A satellite image of the study area acquired on 28 August

2015 after 4 days of non-upwelling winds revealed that surface turbidity at the upwelling area had significantly decreased, however, remained relatively high, as compared to the adjacent sea.

### 3.2. Coastal Upwelling near the Indigirka and Kolyma Deltas in the East-Siberian Sea

Cold and turbid events similar to those observed to the north from the Lena Delta in the Laptev Sea were regularly registered near the large Indigirka and Kolyma deltas in the East-Siberian Sea (Figure 6). We analyzed all MODIS Terra and MODIS Aqua satellite images of the study region taken in 2000–2019 during July–October when the southern part of the East-Siberian Sea was free of ice. We detected 40 and 62 periods when the areas adjacent to the Indigirka and Kolyma deltas, respectively, were free of clouds and the structure of surface turbidity and temperature could be identified.

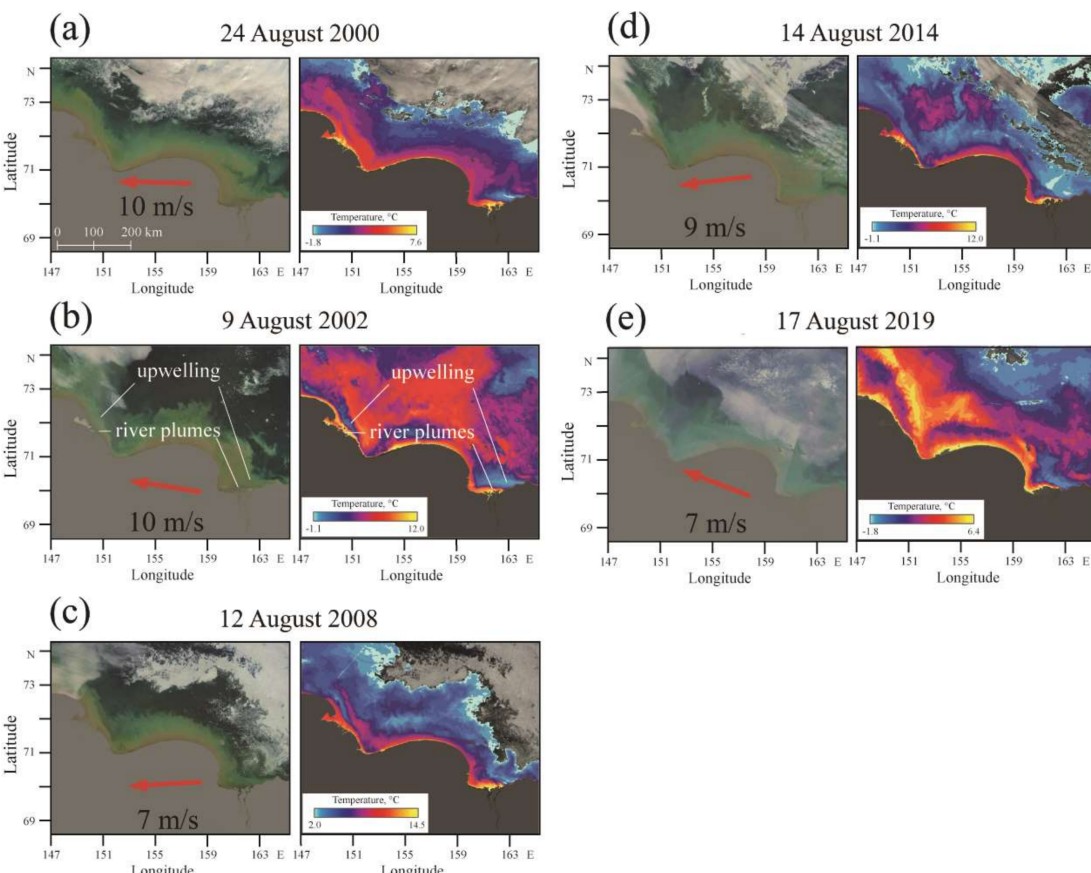

**Figure 6.** Corrected reflectance (left) and brightness temperature (right) from MODIS Terra and MODIS Aqua satellite images of the East-Siberian Sea acquired on (**a**) 24 August 2000, (**b**) 9 August 2002, (**c**) 12 August 2008, (**d**) 14 August 2014 and (**e**) 17 August 2019, indicating location of upwelling events to the north of the Indigirka and Kolyma deltas induced by wind forcing (arrows) and manifested by elevated sea surface turbidity and reduced sea surface temperature. Surface manifestations of upwelling events and river plumes are indicated at panel (**b**).

Similarly to upwelling events near the Lena Delta, the periods of formation of the cold and turbid area near the Indigirka and Kolyma deltas show very good agreement with the periods of upwelling-favorable wind forcing (Figures 7 and 8). The reanalysis wind data reveals that a strong (>6 m/s) easterly and southeasterly wind (100°–160°) for the Indigirka Delta (Figure 7) and strong (>7 m/s) easterly wind (60°–120°) for the Kolyma Delta (Figure 8) were dominating local atmospheric circulation several days before and/or during all periods when the cold and turbid zones were observed on satellite imagery. Black dashed rectangles in Figures 7 and 8 indicate the related upwelling-favorable

conditions near the Indigirka and Kolyma deltas. For all cold and turbid cases, the day of observation (filled red square) and/or the preceding day (empty red square) is located inside these dashed rectangles. On the other hand, the direction of the prevailing wind was different or its velocity was low during all periods when no cold and turbid areas were detected, i.e., all triangles in Figures 7 and 8 are located outside the dashed rectangles. Therefore, we presume that the cold and turbid areas observed to the north of the Indigirka and Kolyma deltas are surface manifestations of wind-driven upwelling events. Similarly to the Lena Delta region, we observe asymmetry in wind forcing conditions with almost absent wind forcing between 135° and 225°.

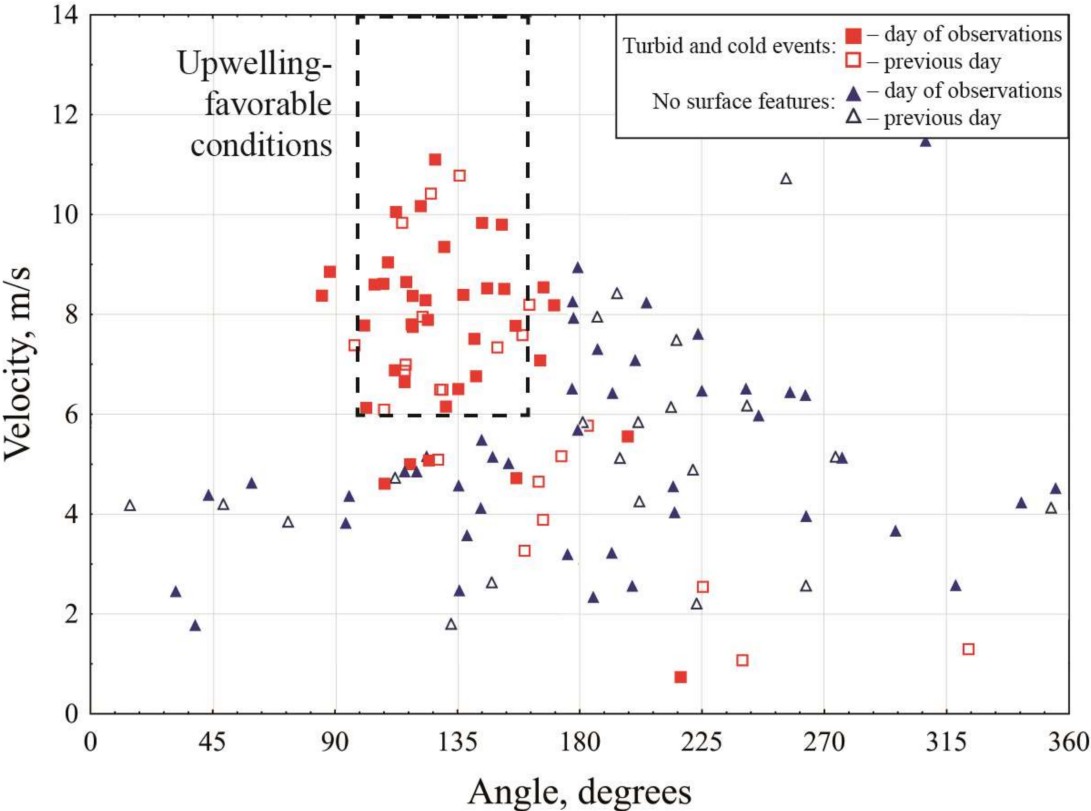

**Figure 7.** Wind forcing conditions at the Indigirka Delta region during periods of presence (red squares) and absence (blue triangles) of cold and turbid events detected on satellite imagery. For each satellite image, averaged wind forcing conditions are shown during the day of satellite observation (filled symbols) and during the preceding day (empty symbols). The black dashed rectangle indicates upwelling-favorable wind forcing conditions.

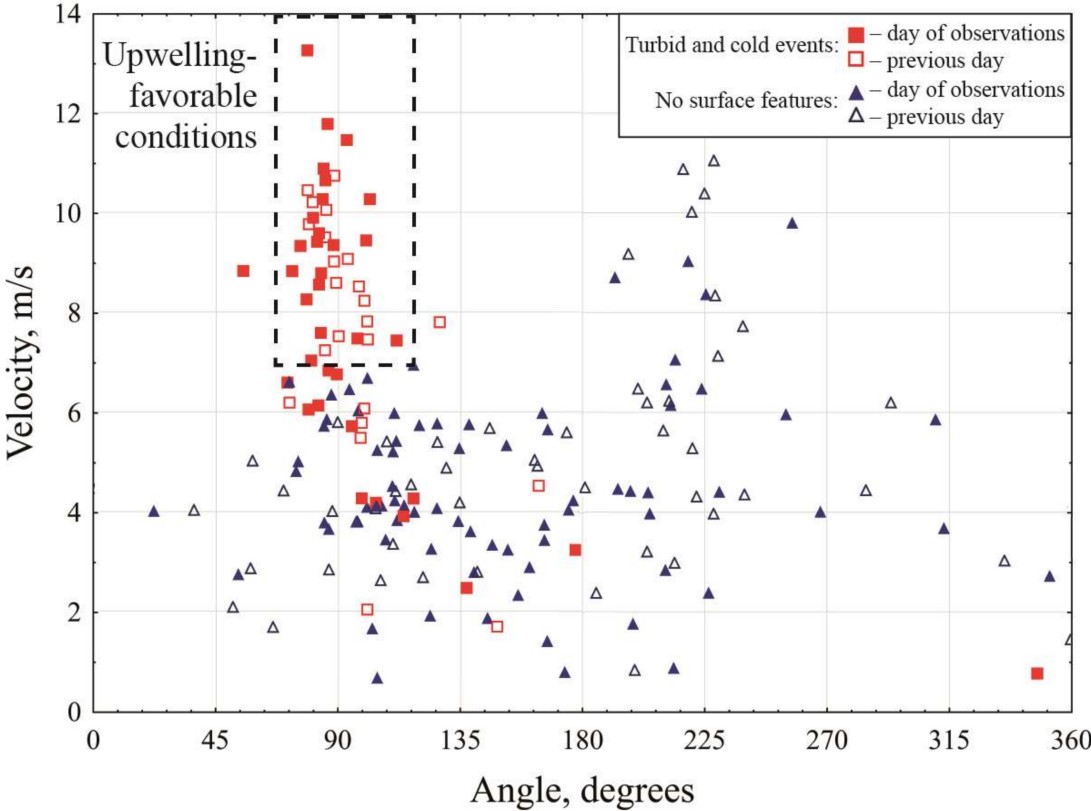

**Figure 8.** Wind forcing conditions at the Kolyma Delta region during periods of presence (red squares) and absence (blue triangles) of cold and turbid events detected on satellite imagery. For each satellite image, averaged wind forcing conditions are shown during the day of satellite observation (filled symbols) and during the preceding day (empty symbols). The black dashed rectangle indicates upwelling-favorable wind forcing conditions.

Upwelling areas in the East-Siberian Sea occupied a large part of the coastal sea adjacent to the Indigirka (3000–6000 km$^2$) and Kolyma (5000–9000 km$^2$) deltas (Figures 6, 9 and 10). Their southern borders are stretched along the northern coasts of the Indigirka and Kolyma deltas. Upwelling events were observed in 21 cases of the 40 considered periods near the Indigirka Delta and in 24 cases of the 62 considered periods near the Kolyma Delta. Similarly to upwelling events near the Lena Delta, we detected the process of development of upwelling events near the Indigirka and Kolyma deltas in response to changes of wind forcing regimes in August 2002 (Figures 9a and 10a), August 2010 (Figure 9b), and August 2014 (Figure 10b). Moderate (2–5 m/s) southeasterly wind forcing was prevailing in the study region on 3–5 August 2002 and its velocity increased to 7–8 m/s on 6 August 2002. No upwelling manifestations were observed on satellite images acquired on 4–5 August 2002 in the study area (Figures 9a and 10a). Then the areas of reduced surface temperature and increased surface turbidity were formed at the isobaths of 5–10 m near the Indigirka and Kolyma deltas on 6 August 2002. These areas increased on 7–10 August 2002, whereby their northern borders steadily propagated offshore, indicating the development of coastal upwelling events in response to strong southeasterly wind (9–10 m/s), which dominated in the study region till 11 August 2002. The upwelling area steadily increased to 6000 km$^2$ near the Indigirka Delta (Figure 9a) and to 9000 km$^2$ near the Kolyma Delta (Figure 10a).

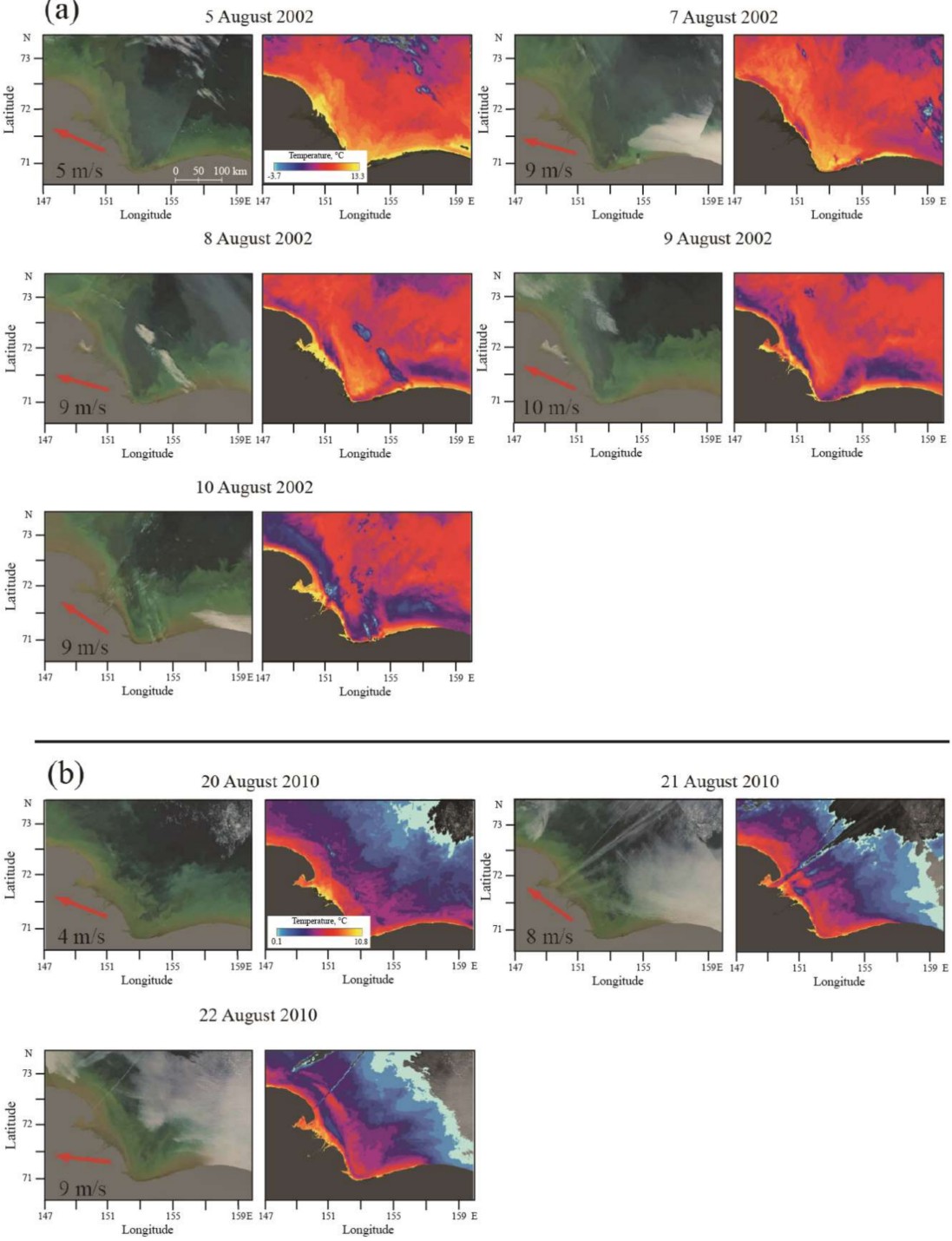

**Figure 9.** Corrected reflectance (left) and brightness temperature (right) from MODIS Terra and MODIS Aqua satellite images of the area adjacent to the Indigirka Delta acquired on (**a**) 5, 7–10 August 2002 and (**b**) 20–22 August 2010 illustrating formation of the upwelling event in response to wind forcing (arrows).

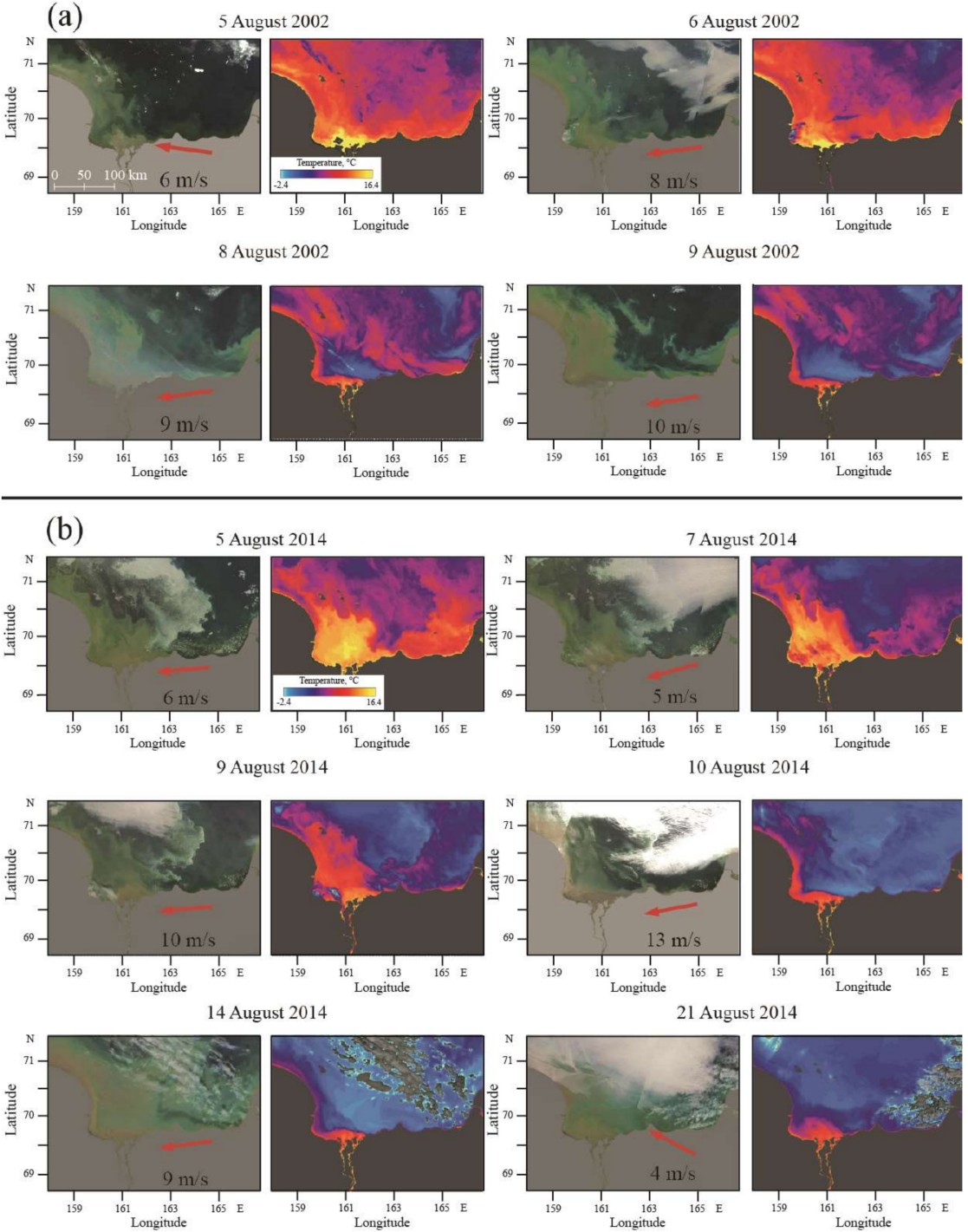

**Figure 10.** Corrected reflectance (left) and brightness temperature (right) from MODIS Terra and MODIS Aqua satellite images of the area adjacent to the Kolyma Delta acquired on (**a**) 5, 6, 8, 9 August 2002 and (**b**) 5, 7, 9, 10, 14, 21 August 2014 illustrating formation and dissipation of upwelling events in response to wind forcing (arrows).

Development of an upwelling event was also registered on 20–22 August 2010 (Figure 9b). No upwelling was observed on 20 August 2010 during moderate (4 m/s) southeasterly wind forcing. Then on 21 August 2010 upwelling wind increased to 8 m/s and formation of a cold and turbid zone started, which is visible on satellite image. The next day, 22 August 2010, a well-developed upwelling event was observed. Formation and dissipation of upwelling near the Kolyma Delta was observed on 5–24

August 2014 (Figure 10b). Satellite imagery show that warm river plume occupied the area adjacent to the Kolyma Delta on 5–8 August 2014 under moderate (4–6 m/s) wind forcing conditions. Coastal upwelling induced by easterly wind (9–13 m/s) on 9–14 August 2014 resulted in mixing of the Kolyma plume manifested by abrupt decrease of surface temperature at the upwelling area. The warm plume remained only in vicinity of the Kolyma Delta, and its area dramatically decreased from 13,000 to 1500 km². Relaxation of upwelling favorable wind (1–5 m/s) on 15–24 August 2014 was accompanied by steady increase of area of the Kolyma plume registered by satellite imagery on 21 and 24 August 2014.

## 4. Discussion

Upwelling winds near the Lena, Indigirka, and Kolyma deltas cause mixing and intense offshore transport of river plumes over sloping seafloor and upward penetration of cold subjacent sea water (Figure 11). The upwelling sea water induces resuspension of bottom sediments and transports them upward to the surface layer. This process strongly depends on local bathymetry, therefore it occurs only over certain zones of the coastal sea. As a result of detachment of river plumes from river delta and upwelling of subjacent sea water, large saline, cold, and turbid "holes" are formed within the Lena, Indigirka, and Kolyma plumes, which are detected on satellite imagery.

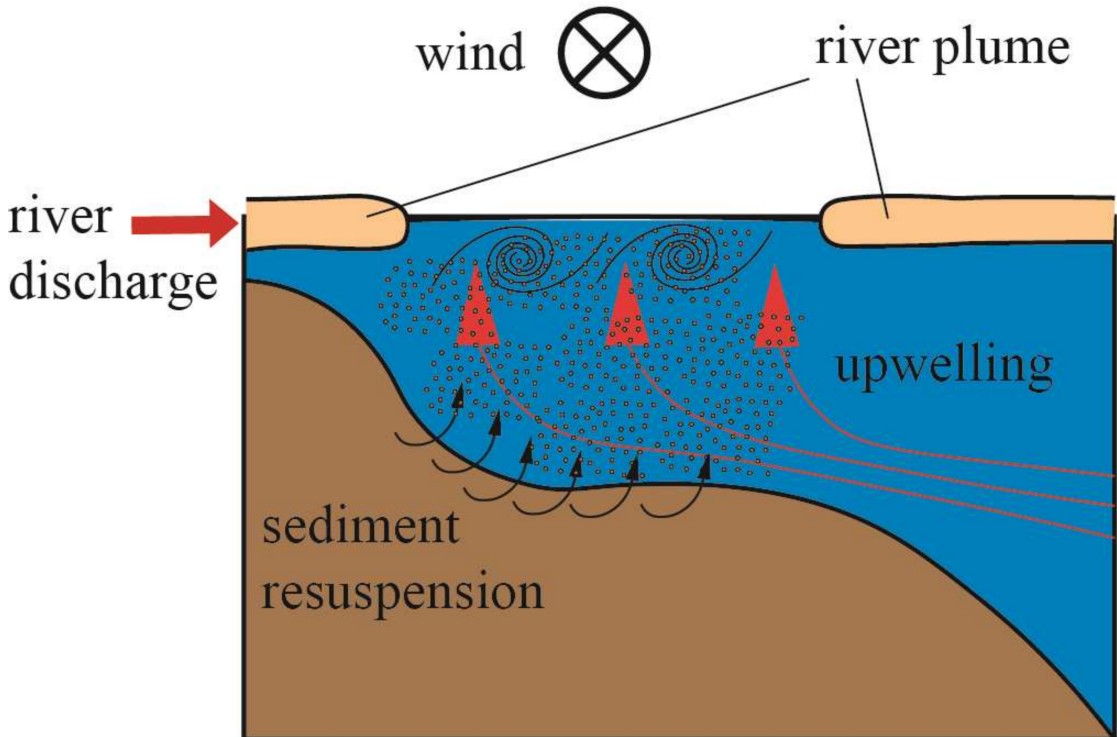

**Figure 11.** Schematic of formation of a saline, cold, and turbid "hole" within a plume as a result of advection and mixing of a river plume during wind-induced upwelling event.

Based on theory described by [39], we quantified the spatial and dynamical characteristics of the response of the Lena, Indigirka, and Kolyma plumes to upwelling-favorable winds. Given the speed of the upwelling wind, we can calculate three key parameters of this process, namely, the depth of the surface layer entrained by offshore displacement $h_s$, the time to separate the plume from the coast $t_{sep}$, and the time to halve the initial salinity anomaly of the plume $t_s$. The first parameter is determined by the equation

$$h_s = \sqrt{\frac{2Ri\rho_{sea}}{gh_p\Delta\rho_p}}U, \tag{1}$$

where $Ri = \frac{g\Delta\rho_p h_p^3}{\rho_{sea} U^2}$ is the Richardson number, $\rho_{sea}$ is the ambient sea density, $g$ is the gravity acceleration, $h_p$ is the plume depth, $\Delta\rho_p$ is the plume salinity anomaly, $U = \frac{\tau}{\rho_{sea} f}$ is the Ekman transport, $f$ is the Coriolis frequency, and $\tau$ is the wind stress. We obtain that if upwelling wind speed exceeds 9 m/s, i.e., $U$ exceeds 1.46 m$^2$, for the Lena plume ($\rho_{sea}$ = 1016 kg/m$^3$, $h_p$ = 5 m, $\Delta\rho_p$ = 4 kg/m$^3$, $f$ = 1.4 × 10$^{-4}$ 1/s, $Ri$ ~ 1 according to [17,27]), then $h_s$ is greater than the plume depth $h_p$, i.e., the whole depth of the Lena plumes is entrained into offshore displacement during an upwelling event. This theoretical estimation of the threshold value for wind speed (9 m/s) is in a good accordance with the threshold value (8 m/s) obtained from analysis of satellite imagery and wind reanalysis described in Section 3.1. Similar assessment of the upwelling wind threshold value for the Indigirka and Kolyma plumes ($h_p$ = 3 m, $\Delta\rho_p$ = 4 kg/m$^3$ according to [17,25]) is equal to 7 m/s, which is also consistent with the threshold values (6 m/s for the Indigirka plume and 7 m/s for the Kolyma plume) reconstructed from satellite imagery and wind reanalysis.

If wind speed exceeds the threshold value, a plume separates from the coast during several hours ($t_{sep}$ ~ 1/$f$ = 1.4 × 10$^4$ s ~ 4 h) and halves its initial salinity anomaly during the time period quantified by the following equation:

$$t_s = \frac{2A_P}{\sqrt{Ri}U},\qquad(2)$$

where $A_p = W_p \times h_p/2$ is the initial cross-sectional area of the plume and $W_p$ is the initial cross-shore extent of the plume. For the considered plumes, we set $W_p$ ~ 5 × 10$^4$–10$^5$ m and obtain $t_s$ = 1.7 × 10$^5$–3.4 × 10$^5$ s ~ 2–4 days for the Lena plume and $t_s$ = 1.5 × 10$^5$–3 × 10$^5$ s ~ 1.5–3.5 days for the Indigirka and Kolyma plumes. As a result, several days of strong upwelling wind are estimated to induce northward offshore displacement of these plumes and halve their salinity anomalies due to intense mixing with subjacent sea. Several days of strong upwelling wind cause formation of $W_p/2$ = 25–50 km wide areas of saline ambient sea water between the plumes and the related deltas that is consistent with satellite observations of the study area (Figures 2 and 6). On the other hand, if wind speeds are smaller than the threshold values, separation of the plumes from the coast occurs after several days or more from the onset of upwelling wind. In this case, mixing of the plumes with ambient sea has low intensity; salinity anomalies of the plumes decrease slowly and halve only after several weeks of upwelling winds [39].

In Section 3 we described ranges of wind speed and wind direction that induce coastal upwelling events at the study regions visible on satellite imagery. We applied these ranges to wind reanalysis and identified periods of wind forcing favorable for formation of upwelling events near the Lena, Indigirka, and Kolyma deltas during the ice-free seasons of 1979–2019 (Figure 12). The average annual duration of upwelling events near the Lena, Indigirka, and Kolyma deltas during this period was equal to 5, 10, and 14 days per year, respectively. The frequencies of upwelling events detected on satellite imagery in the study areas are overestimated by several times, as compared to the obtained average frequencies reconstructed from wind reanalysis. This large bias is caused by, first, detection of residual upwelling events on satellite images that remain during several days after secession of upwelling wind and, second, by almost complete absence of cloud-free satellite imagery during non-upwelling northernly winds described in Section 3.

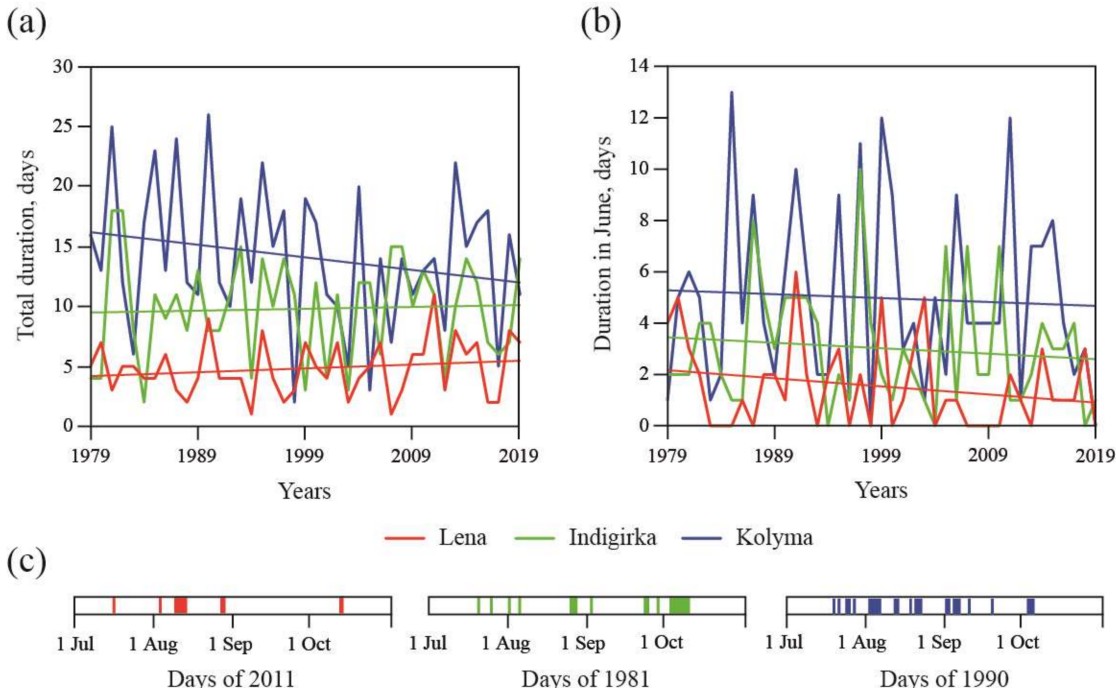

**Figure 12.** (**a**) The total annual duration and (**b**) duration in July of upwelling events near the Lena (red), Indigirka (green), and Kolyma (blue) deltas in 1979–2019. (**c**) Distributions of upwelling periods in July–October in 2011 near the Lena Delta (left), in 1981 near the Indigirka Delta (center), and in 1990 near the Kolyma Delta (right).

The Kolyma River discharge exhibits the longest upwelling-induced mixing, while influence of upwelling events on the Lena River discharge is the smallest among the considered rivers. However, the annual duration of upwelling events near the Kolyma Delta showed a strong negative trend decreasing by 25% from 1979 to 2019 (Figure 12a). The same characteristic for the Lena and Indigirka regions, in contrast, was increasing, albeit less dramatically than at the Kolyma region. The observed trends could be caused by the influence of the ongoing climate change on atmospheric circulation in the Arctic [100–102] and, therefore, on duration of upwelling winds in the study regions.

The annual duration of upwelling events showed substantial inter-annual variability caused by variability of local atmospheric circulation (Figure 12a). It varied from 2–3 days at all study regions to 11 days near the Lena Delta in 2011, 18 days near the Indigirka Delta in 1981 and 1982, and 26 days near the Kolyma Delta in 1990. Therefore, during certain years the wind-induced upwelling events and the related periods of intense mixing of the Lena, Indigirka, and Kolyma plumes account for up to 12%, 19%, and 28% of ice-free periods, respectively. As a result, the total duration of the upwelling periods, which is negligible on an annual scale, is much more significant during certain weeks and months. In particular, the longest registered durations of upwelling events during individual months are equal to 8, 10, and 14 days for the Lena, Indigirka, and Kolyma regions, respectively, i.e., upwelling events occurred during quarter to half of these months.

The diversity of duration of upwelling events in different years and months is illustrated by their durations in July in 1979–2019 (Figure 12b) and by uneven distributions of upwelling periods during the years with their maximal total duration, namely, 2011 for the Lena Delta region, 1981 for the Indigirka Delta region, and 1990 for the Kolyma Delta region (Figure 12c). In order to quantify the inter-annual variability of influence of upwelling events on mixing of freshwater discharge with sea water we analyzed the inter-annual variability of their durations in July (Figure 12b). Freshwater runoff from the Lena, Indigirka, and Kolyma rivers during the end of June and July provides approximately 60% of their total annual discharge and induces melting of sea ice at the areas adjacent to the river

deltas [17]. As a result, long-term upwelling events in July can significantly increase mixing of the river plumes with the subjacent saline sea and strongly affect the structure and dynamics of the related river plumes. Indeed, in certain years upwelling events occurred over 5–6 days in July near the Lena Delta, 7–10 days near the Indigirka Delta, and 10–13 days near the Kolyma Delta. On the contrary, upwelling events and, therefore, upwelling-induced mixing, were completely absent in July during certain years for all three regions. In particular, upwelling events occurred during 0 or 1 day in July near the Lena Delta in 18 out of 41 considered years. As a result, the Lena discharge exhibited negligible upwelling-induced mixing near the delta in July in approximately half of the years during 1979–2019. The reconstructed durations of upwelling events in July showed similar slight negative trends in 1979–2019 at all three considered regions.

## 5. Conclusions

Satellite observations reveal upwelling events that regularly occur during ice-free seasons in the areas adjacent to the Lena, Indigirka, and Kolyma deltas in the Laptev and East-Siberian seas. These areas are manifested by decreased temperature and increased turbidity, as compared to the surrounding sea. Based on meteorological and satellite data, we estimated temporal characteristics of formation and dissipation of these upwelling events in response to variability of wind forcing. Surface manifestations of upwelling events occur after less than 1 day of strong upwelling winds at all three considered regions. Upwelling near the Lena Delta is fully developed and occupied an area of 15,000–17,000 km$^2$ after 4–5 days of strong upwelling-favorable southeasterly winds. Fully developed upwelling events near the Indigirka and Kolyma deltas are formed after 3–4 days of strong upwelling-favorable easterly and southeasterly winds; their areas are 5000–6000 and 8000–9000 km$^2$, respectively. Upwelling areas remain stable until secession of upwelling wind forcing and then steadily dissipate after several days of non-upwelling winds.

The importance of these upwelling events consists in their location near the large river deltas which provide the majority of freshwater discharge to the Laptev (70%) and East-Siberian (75%) seas. Upwelling events induce very intense advection and vertical mixing of freshened surface layer with subjacent saline sea near freshwater sources, as compared to mixing caused by wind-induced shear stress. As a result, the Lena, Indigirka, and Kolyma river plumes are significantly transformed and diluted near their sources during upwelling-favorable wind forcing periods. Frequency and duration of upwelling events govern the structure and dynamical characteristics of the large river plumes, which spread from the river deltas over the upwelling areas to the open sea. Therefore, despite their relatively small areas, upwellings can strongly influence transport and transformation of freshwater discharge over wide areas in the Laptev and East-Siberian seas.

Using NCEP/CFSR/CFSv2 wind reanalysis we reconstructed periods of upwelling events during ice-free seasons at the study areas in 1979–2019. Total annual duration of upwelling events shows large inter-annual variability from negligible (2–3 days) to significant, namely, 12% duration of ice-free periods near the Lena Delta, 19% for the Indigirka Delta, and 28% for the Kolyma Delta. Moreover, upwelling events are unevenly distributed within individual years. In particular, they can last for a quarter to a half of certain months followed by long periods of non-upwelling wind forcing. The most frequent upwelling events among the considered areas are observed near the Kolyma Delta, followed by the Indigirka Delta, and the Lena Delta. As a result, the Kolyma River discharge exhibits the strongest upwelling-induced mixing, however, with strong negative trend registered in 1979–2019. Durations of upwelling events near the Lena and Indigirka deltas, on the other hand, have slight positive trends that decrease their difference with duration of upwelling events near the Kolyma Delta. The revealed trends are presumably caused by long-term changes in atmospheric circulation in the study region induced by the ongoing climate change in the Arctic. Climate change also causes increase of river discharge and temperature of river water, as well as decrease of duration of ice coverage in the coastal areas. Influence of these complex processes on frequency and duration of upwelling events and

intensity of upwelling-induced mixing near the Lena, Indigirka, and Kolyma deltas requires specific research and is a subject of future work.

Coastal upwelling events reported in this study can strongly affect salinity and stratification of the surface layer during ice-free periods and, therefore, influence variability of ice coverage in the Laptev and East-Siberian seas. In particular, enhanced duration and intensity of upwelling-induced mixing activity near the Lena, Indigirka, and Kolyma deltas can increase salinity of the related river plumes and, therefore, decelerate ice formation in the Laptev and East-Siberian seas, as was revealed for the other Arctic seas [103–105]. The considered upwelling events can also strongly influence primary productivity and local food webs. Upwelling causes upward penetration of nutrient-rich sea water [35,43–47,106], which is especially important for nutrient-poor areas at the shelf of the Laptev and East-Siberian seas where vertical convection is inhibited by strong stratification formed by large continental runoff. In particular, elevated concentrations of nitrates and increased biological productivity were reported in the vicinity of the upwelling area located near the Lena Delta shortly after an upwelling event [32]. Therefore, the results obtained in this study hold promise to provide improved assessments of the fate of freshwater discharge in the Laptev and East-Siberian seas, as well as its impact on local physical, biological, and geochemical processes. However, a detailed study of the influence of wind-driven coastal upwelling events on the structure and dynamics of the freshened surface layers in these seas requires specific in situ measurements during upwelling and non-upwelling events, as well as numerical modelling, and is within the scope of future work.

**Author Contributions:** Conceptualization, A.O.; formal analysis, A.O.; investigation, A.O., K.S. and S.M.; writing, A.O., K.S. and S.M. All authors have read and agreed to the published version of the manuscript.

**Funding:** This research was funded by the Ministry of Science and Higher Education of Russia, theme 0149-2019-0003 (collecting of wind reanalysis data); the Russian Foundation for Basic Research, research projects 18-05-60069 (collecting of satellite data), 20-35-70039 (analysis of satellite data), and 18-05-00019 (study of river plumes); the Russian Science Foundation, research project 18-17-00089 (analysis of wind reanalysis data); the Grant of the President of the Russian Federation for state support of young Russian scientists—candidates of science, research project MK-98.2020.5 (study of freshwater transport).

**Acknowledgments:** This research benefited substantially from fruitful discussions with Natalia Tilinina (Shirshov Institute of Oceanology). The authors are grateful to the editor Esther Liu and two anonymous reviewers for their comments and recommendations that served to improve the article.

**Conflicts of Interest:** The authors declare no conflicts of interest.

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
