# Peer review of "Wind-Driven Coastal Upwelling near Large River Deltas in the Laptev and East-Siberian Seas"

_remotesensing, doi:10.3390/rs12050844_

Round 1

Reviewer 1 Report

Wind-Driven Coastal Upwelling Near Large River Deltas in the Laptev and East-Siberian Seas

Abstract:

It is correct.

Keywords:

Possibly there are too many keywords, they could be reduced by half or six.

Introduction:

It is correct.

Study Area and Data:

Suggestion:

The authors have divided the sections into:

Study area and data.

Results.

Change is not mandatory, but I understand that division is better:

Study area

Data and results

The data collection does not deserve a special section, since it is very simple, but the data collection with the results may be better placed. Better leave only the study area section.

Study area and data are correct. But Figure 1a would put a graphic scale, to detail the distance existing in the various study.

Results:

Figures 2, 3, 6, 9 and 10 could have a color scale that indicated the thermal value of the representation of the images.

Figure 3: What is the reason that the figures do not follow a temporal order ?: a) August 25, 2000. b) September 11, 2005. c) August 3, 2019.

Figure 4, 7 and 8: A chart can be made indicating the meaning of the symbology, instead of explaining it at the bottom of the figure.

Figure 4: Almost all red squares (wind presence) are between 300 degrees and 360 degrees and none have a value greater than 0 degrees. It could happen that situations of red squares in values from 0 to 50 degrees as a continuation of 360 degrees. What I indicate if it occurs with the triangles (absence of wind) that are distributed between 300 and 135 degrees.

It is striking that there are no wind directions between 130 and 300 degrees, this is north-south direction. Are the winds always south-north?

Discussion:

There are large paragraphs of results that could be brought to the discussion section. The dicussion of the results is developed in the results section when they should be brought to the discussion section.

Formulas 1 and 2 do not look well written. Review mathematical expressions of the text, such as: (111 ∙). In Adobe (PDF) the expressions work out well but in Word (DOC) they are not correct.

Conclusions:

Figure 11 would avoid putting it at the end of the text and it has too large text with respect to the text.

The turbidity of the water is affected by the internal storms of the river, this is the flow that can take after events of heavy rains. Possibly, this amount of water affects the temperature of the mouth. What do you think are the reasons that make turbidity and temperature vary? Can climate change affect ?, is there a correspondence between hot years with the highest temperature of end of the river?.

The study is based on the information provided by satellite images, but have these values been contrasted for the same dates with those obtained at the site, with their own information?.

Reviewer 2 Report

The authors have studied the formation and dissipation of coastal upwelling near three large rivers flowing into the Laptev and East-Siberian Seas in response to wind forcing. They have done the survey using corrected reflectance and brightness temperature retrieved from Terra MODIS and Aqua MODIS satellite imagery as well as the NCEP/CFSR reanalysis data. The research has extended the knowledge on drivers and occurrence of coastal upwelling in the Laptev and East-Siberian Seas, as well as the impact of coastal upwelling on the spreading and mixing of fresh waters in those reservoirs. Specific comments are given below.

Page 1, line 17 + plus many other examples: Giving an information on the direction of wind, we give the information from which direction the wind is blowing. Wind blowing from the north is recorded as 0 degrees. But when we give an information on sea currents we give information in which direction the current is flowing. It is customarily accepted. Beginning from abstract through all the ms, the authors were using: “westward”, “northwestward”, etc., while they were writing about winds. It is really confusing, especially when degrees are given. Please correct all the ms.

Page 3, Figure 1a: Whereas Figure 1b is clear and well prepared, Figure 1a should be corrected. The shoreline is fuzzy. Indicate rivers flowing into seas being under consideration.

Page 4, lines 104-113: Describe the satellite data used in the study in detail. How maps of sea surface distributions of corrected reflectance and brightness temperature were retrieved?

Page 4, lines 112-113: Should “Corrected Reflectance and Brightness Temperature” be written with capitals?

Page 4, line 116: There is no chapter on methods used in the study. The authors did not present any method of upwelling detection. Instead of it, the authors writing about upwelling phenomena used expressions like “increased sea surface turbidity and reduced sea surface temperature” or “turbid and cold events”, Please quantify ‘increased sea surface turbidity and reduced sea surface temperature’ and “turbid and cold events”. Moreover, in all the ms there is no information given on sea surface temperature of upwelling waters and adjacent areas as well as on turbidity. 

Page 4, lines 123-125: Please quantify when you decided to qualify an upwelling event as “distinct”.

Page 4, lines 126-137: Please indicate in the Figure 2 areas influenced by an upwelling and a river discharge.

Page 4, lines 138-153: The information on processes occurring in the seas under consideration, among others on characteristics of coastal upwelling should be given in introduction.

Page 5, line 154: Add scales, units, geographical coordinates into Figure 2. Indicate places of upwelling.  Divide Figure 2 on two figures: corrected reflectance and brightness temperature… or number consecutively all the pictures in Figure 2. Now one letter is always for two pictures.

Page 6, Figure 3: Add scales and units into Figure 3. Number consecutively all the pictures in Figure 3. Now one letter is always for two pictures. Otherwise you can use “left image” and “right image” expressions for figures presenting corrected reflectance and brightness temperature respectively.

Page 7, Figure 4: As I have written at the beginning of the review, it is really confusing not explaining wind conditions writing from which direction the wind is blowing. Redraw the figure - OX axis: degrees showing from which direction the wind is blowing.

Page 7, line 196: Should “World” be written with a capital?

Page 8, line 209 – “strong southeastern winds” should be. Correct all the ms, also where degrees are given (for example in line 210).

Page 8, Figure 5 – add the scale and units.

Page 9, line 234 + plus many other examples – quantify “turbid and cold…”

Page 9, line 246 – as I stated before – “a week of strong eastern winds…” should be. Correct all the ms.

Page 9, line 247 + plus many other examples – quantify “distinct”

Page 10, Figure 6 – add scales, units, geographical coordinates, number consecutively all the pictures in Figure 6, indicate areas of coastal upwelling and influenced by fresh waters.

Page 11, Figure 7 and page 12, Figure 8 – the same as for Figure 4.

Page 12, Figure 9 and Page 13/14, Figure 10 - add scales, units, geographical coordinates, number consecutively all the pictures in Figures 9 and 10, indicate areas of coastal upwelling.

Page 14, lines 330-339 – The authors stated that they estimated temporal characteristics of formation and dissipation of upwelling events but… they didn’t present methods of upwelling detection. Moreover they forgot about quantitative description of results. Even figures have no scales and units.

Page 15, line 360 – In my view, theoretical explanations should be in the chapter devoted to methods.

Page 15, lines 373-385 – it’s a very interesting paragraph. Could you broaden it presenting some more information about frequency of upwellings from NCEP/CFSR wind reanalysis and satellite data? Are numbers of days presented in the line 378 calculated for all the year (like it is written) or for July-October. Could you compare those frequencies with calculated from satellite images?

Page 15, line 382 – Instead of writing “during certain years…” it’s better to show diversity of upwelling events and the related periods of intense mixing in different years and months in pictures.

Page 16 – the chapter Conclusions should be devoted to conclusions drawn from this study on coastal upwelling and its interactions with river discharge in the Laptev and East-Siberian Seas, and should highlight findings. A few sentences about future work could be useful as well. Instead of conclusions in this ms the authors inserted a figure and the explanation of the figure. Please move it with the explanation to the discussion and write conclusions. Moreover the quality of Figure 11b is bad, and in my view the location of main rivers should be presented at the beginning of the ms, so in Figure 1. The figure is fuzzy. Do values with units km3 represent an annual mean river flow? It isn’t written. Are blue patches upwelling areas? Do red arrows indicate river mouths?  

Reviewer 3 Report

The manuscript deals with coastal upwelling in the Arctic Sea off-shore the Lena, Kolyma and Indigirka rivers. The study is based on a combined analysis of satellite observations of turbidity and brightness combined with meteorological reanalysis data. It is a very interesting study that makes excellent use of satellite observations in these remote areas where land/sea observations are scarse. The manuscript is clear and written in a good English. I have a few comments that I ask the authors to consider.

Major comment:

  1. Line 205 and throughout the whole paper incl. Figs 4, 7 and 8, the wind direction does not follow the meteorological standard – which is confusing and hampers the readability of the paper. The indication of wind direction in the manuscript should be such that e. g. a westerly wind – 270 degrees – comes from west and blows towards east. In the paper the wind direction is shifted 180 degrees such that a westerly is names eastwards – 90 degrees. This must be changed throughout the whole manuscript.
  2. Line 109 on data. Add some information on turbidity and how it actually is estimated (basically water color) and how accurate the estimate is believed to be– actually what is it from a remote sensing point of view. The same applies for the brightness temperature.
  3. Figs 2, 3, 5 , 6, 9 and 10. Add color bars to the plots that give an indication of the connection between temperature and color/turbidity and color. By adding these color bars it should be easy to identify cold areas and high turbidity areas respectively.

Minor comments:

  1. Line 346. Define the Ri-number.
  2. Line 349: Discuss why Ri is taken to be about 1.

Round 2

Reviewer 2 Report

The ms, including the text and figures, was corrected and rewritten according to my suggestions. The authors did a really good job.

Author Response

Many thanks for your valuable comments and suggestions that served to improve the article. In the revised version of the manuscript we performed final spell check and corrected a few mistakes.